# MODALITY-SWAP DISTILLATION: RENDERING TEXTUAL REASONING INTO VISUAL SUPERVISION

## ABSTRACT

Visual reasoning over structured data such as tables is a critical capability for modern vision-language models (VLMs), yet current benchmarks remain limited in scale, diversity, or reasoning depth, especially when it comes to rendered table images. Addressing this gap, we introduce **Visual-TableQA**, a large-scale, open-domain multimodal dataset specifically designed to evaluate and enhance visual reasoning over complex tabular data. Our generation pipeline is **modular, scalable, and fully autonomous**, involving multiple reasoning LLMs collaborating across distinct roles: generation, validation, and inspiration. **Visual-TableQA** comprises 2.5k richly structured LaTeX-rendered tables and 9k reasoning-intensive QA pairs, all produced at a cost of under $100. To promote diversity and creativity, our pipeline performs **multi-model collaborative data generation** via **cross-model prompting ('inspiration')** and LLM-jury filtering. Stronger models seed layouts and topics that weaker models elaborate, collectively distilling diverse reasoning patterns and visual structures into the dataset. Empirical results show that models fine-tuned on **Visual-TableQA** generalize robustly to external benchmarks, outperforming several proprietary models despite the dataset's synthetic nature. The full pipeline and resources are publicly available in our GitHub repository.

## 1 INTRODUCTION

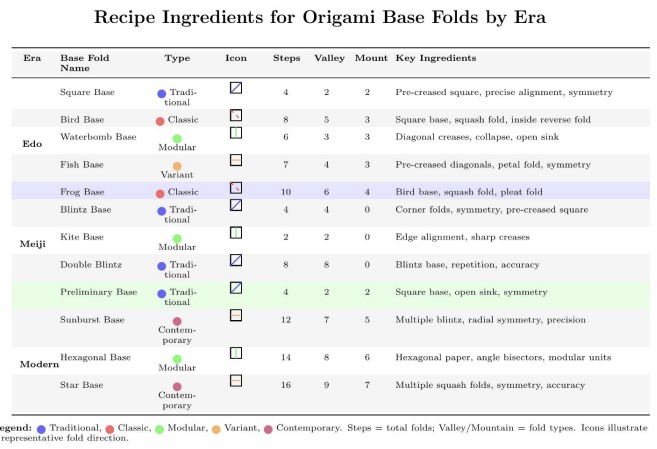

Q: Based on the color coding and icon patterns, which base fold in the Edo era is most likely to involve a combination of both diagonal and straight folding techniques?

A:**Reasoning:** The Fish Base in the Edo era is colored orange (basecolor{4}), which is distinct from the other Edo era bases. Its icon is oldicon{4}, which draws a horizontal line, suggesting a straight fold. However, the Fish Base also involves precreased diagonals and petal folds, which are diagonal techniques. The combination of these elements indicates a mix of diagonal and straight folding. **Conclusion:** The answer is Fish Base from the Edo era.

Figure 1: Sample question in our benchmark

Vision-language models (VLMs) have significantly advanced in recent years, achieving remarkable performance in various tasks involving visual and textual inputs. Despite these advancements, complex reasoning tasks, especially those requiring deep comprehension of tabular data structures, continue to pose significant challenges. Table complexity can manifest in various ways, including structural layout, information density, and the diversity of visual components such as the integration of diagrams. The more complex a table is, the more it lends itself to challenging reasoning tasks,

requiring advanced cognitive abilities to extract relevant information and perform multi-step logical analysis. For example, the table in Figure 1 exemplifies this complexity through its use of multi-row cells, integrated diagrams, and color encoding. Answering the question requires the VLM to interpret information across all cells and perform a sequence of reasoning steps.

Existing table-based QA datasets predominantly fall into two categories: *(i)* those represented purely in textual format—such as WikiTableQuestions Pasupat & Liang (2015), HybridQA Chen et al. (2020b), and AIT-QA Katsis et al. (2022)—which bypass the challenges of visual layout interpretation; and *(ii)* those that lack diversity in visual layouts, visual complexity, and reasoning depth due to being domain-specific (e.g., TAT-DQA Zhu et al. (2022)), or having standardized queries (e.g., TableVQA-Bench Kim et al. (2024)), or highly technical in nature (e.g., Table-VQA Tom Agonnoude (2024)). This second datasets category typically rely on a limited set of layout templates and involve relatively simple visual tasks or basic QA scenarios, falling short of the complexity required for thorough evaluation and advancement of reasoning capabilities. More recent efforts—such as ChartQA Masry et al., ReachQA He et al., and MATH-Vision Wang et al. (2024b)—have aimed to address the need for open-domain coverage, incorporating more diverse visual features, varied question types, and deeper reasoning challenges. However, these datasets primarily focus on charts and function plots, overlooking tables—and with them, an entire dimension of informational structure and layout diversity. An extensive comparison of diverse chart and table datasets is provided in Table 1.

Inspired by ReachQA's Code-as-Intermediary Translation (CIT)—a technique that translates chart images into textual representations while faithfully preserving visual features—we introduce **Visual-TableQA**, a novel synthetic, multimodal, and open-domain dataset tailored to enhance reasoning capabilities through complex table-based question-answering tasks. Visual-TableQA capitalizes on the ability of reasoning-oriented LLMs to generate intricate LaTeX tables, thus significantly reducing costs and eliminating the need for extensive manual annotations. This modality-swap makes it possible for LLMs to invest their textual reasoning ability into visual image in order to improve visual understanding and reasoning. Visual-TableQA emphasizes structural reasoning over domain knowledge. Each entry couples a rendered table image with a complex, visually grounded reasoning task. Tasks require interpreting visual layout cues such as cell alignment, hierarchical headers, merged cells, or embedded symbolic content—emulating real-world documents where visual context is essential for correct interpretation. The dataset contains 2.5k reasoning-intensive tables and 9k QA pairs crafted to assess both information extraction and multi-step reasoning capabilities, all generated at a cost of under $100. The entire dataset has been validated using a committee of high-performing reasoning LLMs, the ROSCOE step by step reasoning score Golovneva et al., and a sample of 800 QA pairs has undergone manual verification by human annotators. In contrast to previous synthetic datasets, Visual-TableQA is less guided in its generation process, allowing for more diversity and creativity in both table complexity (e.g., structural layout, information density, visual component variety) and the design of QA pairs explicitly crafted to challenge visual reasoning skills. We evaluated a broad range of VLMs, from lightweight models to state-of-the-art architectures, and benchmarked their performance against existing datasets. The results show that most VLMs continue to struggle with table understanding.

In sum, our main contributions are:*(i)* a high-quality, visually diverse, and open-domain dataset for table-based reasoning; *(ii)* an LLM-driven, low-cost generation pipeline using cross-model inspiration; *(iii)* an empirical analysis comparing Visual-TableQA to existing table and chart datasets; *(iv)* an extensive evaluation of open and proprietary VLMs, showing performance gains after finetuning. Our dataset and code are publicly available at `https://github.com/AI-4-Everyone/Visual-TableQA`.

## 2 VISUAL-TABLEQA DATASET

Unlike previous datasets that rely heavily on textual input or handcrafted annotations, Visual-TableQA leverages a scalable generation pipeline rooted in LaTeX-rendered table images, automated reasoning task creation, and LLM-based evaluation. This strategy enables high diversity and reasoning depth while keeping annotation costs minimal, totaling under $100 using a combination of open-access APIs and limited usage tiers. In this section, we describe our LaTeX-based table encoding 2.1, the data generation pipeline 2.2, and the quality assurance process 2.3.

Table 1: Comparison of existing chart and table datasets across data, Q&A, and dataset properties. Abbreviations: Repr=Representation, Vis= Visual, Comp= Complexity, Temp = Template, Refer = Reference, Rat = Rational, Synth= Synthetic, Scal = Scalable. Cells marked with ▲indicate mixed attributes (e.g., partially template-based; scalable Q&A but non-scalable chart data)

| Datasets | Data Properties | | | | Q&A Properties | | | Dataset Properties | | |
|---|---|---|---|---|---|---|---|---|---|---|
| | # Layouts/ # Topics | Type | Data Repr. | Vis. Comp. | Temp. Free | Vis. Re-fer. | Rat. An-not. | Synth. | #Samples / #QA | Scal. |
| WikiTableQuestions (Pasupat & Liang, 2015) | – | Table | Text | ✗ | ✗ | ✗ | ✗ | ✗ | 2.1k/22k | ✗ |
| HybridQA (Chen et al., 2020b) | – | Table | Text | ✗ | ✓ | ✗ | ✗ | ✗ | 13k/70k | ✗ |
| AIT-QA (Katsis et al., 2022) | –/1 | Table | Text | ✗ | ✓ | ✗ | ✗ | ✗ | 116/515 | ✗ |
| TAT-DQA (Zhu et al., 2022) | –/1 | ▲ | Image | ✗ | ✓ | ✓ | ✓ | ✗ | 2.5k/16.5k | ✗ |
| Table-VQA (Tom Agonnoude, 2024) | –/– | Table | Image | ✗ | ✓ | ✓ | ✓ | ✓ | 16.4k/82.3k | – |
| TableVQA-Bench (Kim et al., 2024) | 11/4 | Table | Image | ✗ | ✓ | ✓ | ✗ | ▲ | 894/1.5k | ▲ |
| ChartQA (Masry et al.) | 3/15 | Chart | Image | ✗ | ✓ | ✓ | ✗ | ✗ | 21.9k/32.7k | ✗ |
| DocVQA (Mathew et al., 2020) | 20/5 | ▲ | Image | ✗ | ✓ | ✓ | ✗ | ✗ | 12.7k/50k | ✗ |
| MultiModalQA (Talmor et al., 2021) | 16/∞ | ▲ | Image | ✗ | ✗ | ✓ | ✗ | ▲ | 29,918 | ✗ |
| MATH-Vision (Wang et al., 2024b) | –/16 | ▲ | Image | ✓ | ✓ | ✓ | ✗ | ✗ | 3k/3k | ✗ |
| REACHQA (He et al.) | 32/∞ | Chart | Image | ✓ | ✓ | ✓ | ✓ | ✓ | 3.7k/22k | ✓ |
| **Visual-TableQA (ours)** | /∞ | Table | Image | ✓ | ✓ | ✓ | ✓ | ✓ | 2.5k/ 9k | ✓ |

Table 2: Model performance on the test sets of four benchmarks: ChartQA, ReachQA, MATH-Vision, Visual-TableQA, and Visual-TableQA-CIT. Visual-TableQA-CIT is the variant of our dataset where tables are represented in LaTeX code form rather than as rendered images. The ReachQA score is reported as the average across its two evaluation splits: *Reasoning* and *Recognition*. The values in blue are from our own evaluation using the LLM jury, while the remaining values are taken from model authors or official leaderboards/model cards. When a fine-tuned model achieves better performance, the result is annotated with ↑; if the performance worsens, it is marked with ↓. The best performance for each model variants and task is in **bold**.

| Models | ChartQA | ReachQA | MATH-Vision$_{FULL}$ | Visual-TableQA | Visual-TableQA-CIT |
|---|---|---|---|---|---|
| | | | *Baseline* | | |
| Human | – | 74.85 | 68.82 | – | – |
| | | | *Proprietary VLMs* | | |
| GPT-4o | 85.7 | 53.25 | 30.39 | 78.5 | 91.0 |
| GPT-4o mini | 77.52 | 40.35 | 28.85 | 67.0 | 80.27 |
| Gemini 2.5 Flash | 84.64 | 56.97 | 41.3 | 85.72 | 92.3 |
| Gemini 2.5 Pro | 85.73 | 61.87 | **73.3** | **86.63** | 86.27 |
| Claude 3.5 Sonnet | **90.8**$^*$ | **63** | 32.76 | 82.46 | **88.5** |
| | | | *Open-Source VLMs* | | |
| Llama 4 Maverick 17B-128E Instruct | 85.3$^*$ | 47.98 | **45.89** | **80.75** | 87.0 |
| Mistral Small 3.1 24B Instruct | 86.24$^*$ | 42.45 | 32.45 | 73.2 | 80.25 |
| Qwen2.5-VL-32B-Instruct | 79.75 | **49.5** | 38.1 | 80.45 | **81.93** |
| Qwen2.5-VL-7B-Instruct | 87.3$^*$ | 49.23 | 25.1 | 71.35 | – |
| | | | *Finetuned VLMs* | | |
| Qwen2.5-VL-7B-Instruct + Visual-TableQA | **84.52**↑ | **60.95** ↑ | **49.77** ↑ | **82.98** ↑ | N/A |
| Qwen2.5-VL-7B-Instruct + ReachQA | 77.59 ↓ | 55.75 ↑ | 48.57 ↑ | 60.68 (56.13) ↓ | N/A |

$^*$ Performance metrics are measured using *Relaxed Accuracy*, which allows for small numerical deviations in the predicted answers. We assume that this accuracy inflates the actual accuracy by at least 5%. This margin is subtracted when selecting the best-performing results, which are shown in **bold**.

## 2.1 MODALITY-SWAP: TABLE REPRESENTATION IN LaTeX

Our approach is inspired by He et al., which demonstrated that state-of-the-art VLMs can reason about visual content even in the absence of explicit visual input. Building on this insight, and leveraging the strong coding capabilities of reasoning-oriented language models across multiple programming languages, we chose to use an intermediate representation of tables in LaTeX rather than directly generating rendered table images. This strategy enables the generation of complex visual tables as compact LaTeX code—typically around 100 lines per table—drastically reducing the cost of generation by minimizing the number of output tokens required in API calls. We refer to this as modality-swap: LLMs invest their textual reasoning ability (programming and analysing programs) into visual images in order to improve visual understanding and reasoning. An ablation study of the modality-swap assumption is reported in Table 2. We observe that most state-of-the-art models perform better when the same query is provided in textual form rather than as a visual input. This suggests both room for improvement in visual reasoning and the potential for transferring textual reasoning capabilities to the visual modality.

Our observations (Table 3) align with those of Kale & Nadadur (2025), who reported that LLMs struggle with LaTeX generation—particularly as task complexity increases, leading to a notable drop in accuracy. Table 3 presents the performance of various models in generating LaTeX tables that compile without errors.

Table 3: Percentage of successful LaTeX compilations for various models. Each accuracy is computed from at least 500 generated samples. The Adjust column indicates the level of manual correction needed to make the table look good: Low means minimal or no adjustments, Medium corresponds to 3–5 required fixes, and High indicates more than 5 adjustments were necessary. The tables generated by DeepSeek-R1-Distill-Qwen-32B never compiled.

| Model | Acc. (%) | Adjust | Model | Acc. (%) | Adjust |
|---|---|---|---|---|---|
| Llama 4 Maverick 17B-128E Instruct Meta AI (2025) | 69 | High | DeepSeek-R1-Distill-Qwen-32B DeepSeek-AI (2025) | 0.0 | – |
| Gemini 2.0 Flash Google (2025a) | 65.7 | Low | DeepSeek-R1T-Chimera TNG Technology Consulting GmbH (2025) | 43.4 | Medium |
| Gemini 2.5 Flash Google (2025b) | 43 | Medium | Claude Sonnet 4 Anthropic (2025) | 56 | Low |
| Gemini 2.5 Pro Google (2025c) | 19.6 | Low | Claude 3.5 Haiku Anthropic (2024) | 64.4 | Low |
| GPT-4.1 OpenAI (2025a) | 41.5 | Low | Grok 3 Beta xAI (2025) | 47.3 | Low |
| Qwen3-30B-A3B Qwen Team (2025a) | 69.4 | Low | Reka Flash 3 Reka AI (2025) | 19.3 | Medium |
| Qwen-QwQ-32B Qwen Team (2025b) | 38.2 | Low | | | |

## 2.2 DATA GENERATION PIPELINE

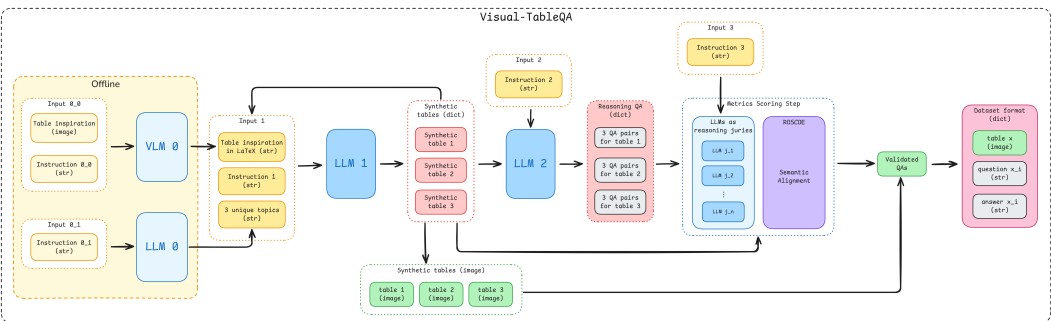

Figure 2: Overview of the full pipeline architecture of Visual-TableQA. A subset of initial table images is first converted to LaTeX using a visual language model (VLM-0). The resulting LaTeX code, along with topic prompts, is then passed to a language model (LLM-1) to generate new, diverse tables. These newly generated tables are used as inputs for further iterations of table generation. All generated tables are then submitted to a second language model (LLM-2), which produces corresponding question-answer pairs. Finally, the QA pairs are evaluated by a jury of high-performing LLMs, and their quality is assessed using the ROSCOE score.

This section provides a detailed description of the generation pipeline. Figure 2 gives an overview of the whole process.

**Seed Tables and Topics Collection:** The first step involves collecting a diverse set of table layouts to serve as inspiration for LLMs during the generation process. We explored various sources, including scientific journals, financial report databases, online newspapers, and table design galleries. Our search included both table and diagram images to introduce greater visual and structural complexity into the dataset. We selected 20 representative images (Figure 6a) and passed them to a visual language model, VLM-0 (`GPT-o3` OpenAI (2025)), to generate accurate LaTeX representations. In parallel, we used LLM-0 (`GPT-4o` OpenAI (2024)) to generate a list of 5,000 distinct topic prompts. These initial table samples and topics serve as the first layer of inspiration for subsequent LLM generations—though the pool of inspirations expands automatically, as detailed in Section 2.2. For reproducibility, all resources are publicly available in our GitHub repository.

**Table Generation:** For each iteration, we randomly select an LLM-1 from the models short-list presented in Table 3. The model receives one table sample from our pool and three topics randomly selected from the topic list, all delivered through a single instruction prompt. The output from LLM-1 is returned as a JSON file containing three newly generated LaTeX-formatted tables in plain text, each corresponding to one of the provided topics. We require that the generated tables be inspired by the input table but include substantial layout variations and, when appropriate, additional data to enhance complexity. The resulting LaTeX code is then compiled using standard LaTeX compilation stack (pdflatex + pdf2image), and cropped to produce high-resolution table images. A human reviewer then inspects the table and makes adjustments to the LaTeX code if necessary. The prompt used for generation are provided in Figure 7.

**Evolving Layouts through Iterations:** A subset of the generated tables is manually selected to enrich the pool of table inspirations. This feedback loop encourages the emergence of increasingly complex and diverse layouts by amplifying visual variations and enabling cross-model inspiration across different LLM-1s over successive iterations. This process is facilitated by the fact that LLMs differ in architecture and tend to focus on distinct structural and stylistic aspects of tables. As a result, combining inspirations across models leads to highly diversified and creative layout types. We refer to this phenomenon as **cross-model prompting ('inspiration')**.

**QA Generation:** Next, for each generated table, we randomly select a model, denoted LLM-2, from the same list of models in Table 3 to generate three QA pairs. The model receives the table in LaTeX format and is instructed to produce questions that require multi-step reasoning, pattern recognition, and symbolic interpretation. For instance, the sample in Figure 1 illustrates how the questions extend beyond basic information extraction, requiring interpretative reasoning to identify patterns within the presented data. We do not fact-check the generated tables; as a result, some table content may be non-factual. While this is important to consider when using the dataset for training, it can be beneficial, as it encourages models to rely on reasoning rather than prior knowledge.

## 2.3 QUALITY ASSURANCE

To ensure the validity of the tables and QA pairs, a panel of independent LLMs—serving as a reasoning jury—evaluates each table and its associated QA pairs by providing binary correctness judgments. The evaluation is based on four criteria: *(i)* the generated document is a valid table and is relevant to the given topic; *(ii)* the table and any associated figures are coherent and meaningful; *(iii)* the question is fully grounded in the table, requiring no external knowledge; and *(iv)* the answer is completely supported by the table content. If any of these four criteria are not met, the corresponding table and its QA pairs are discarded. The LLM jury includes `Mistral-large`, `Deepseek-v3.1`, `Gemini-2.5-pro`, `GPT-4.1`, and `Deepcogito-v2`—models chosen for their strong reasoning abilities. Final acceptance is determined via majority vote across the jury. The prompt used is provided in Figure 9.

The next step involved computing the ROSCOE reasoning scores as introduced in Golovneva et al.. These metrics assess the coherence, logical soundness, and contextual grounding of step-by-step generated rationales. The ROSCOE framework encompasses thirteen evaluation criteria, which we report in Table 7 along with their corresponding values computed over our dataset. The results indicate near-perfect alignment with the expected directionality of each metric, supporting the overall quality of the generated reasoning chains.

**Test Set Construction and Human Evaluation:** The dataset was divided into three subsets: training, validation, and testing. To prevent data leakage, all entries {table, question, answer} derived from a single table were assigned to the same subset. The testing set was also used for human evaluation. Two human annotators—each holding at least a Master's degree and with prior experience in data annotation—were hired to evaluate the quality of 800 QA pairs. Each QA pair was assessed for validity and rated on a scale from 1 to 5. Overall, 92% of the evaluated QA pairs received a rating of at least 4 stars from both annotators.

## 3 EXPERIMENTS

### 3.1 BENCHMARK COMPARISON

**Evaluated Benchmarks and Model Selection:** We evaluate a range of state-of-the-art reasoning VLMs on Visual-TableQA and compare their performance across three other benchmarks focused on table and chart-based visual question answering: ChartQA Masry et al., ReachQA He et al., and MATH-Vision Wang et al. (2024b). Our model selection includes powerful proprietary models such as `GPT-4o`, `GPT-4o Mini` OpenAI (2025b), `Gemini 2.5 Flash`, `Gemini 2.5 Pro`, and `Claude 3.5 Sonnet`, as well as open-source models like `LLaMA 4 Maverick 17B-128E Instruct`, `Mistral Small 3.1 24B Instruct` Mistral AI (2025), `Qwen2.5-VL-32B-Instruct` Chen et al. (2024a), `Qwen2.5-VL-7B-Instruct` Team (2025), `LLaVA-Next-Llama3-8B` Li et al. (2024), `MiniCPM-V2.5-Llama3` Yao et al. (2024), and `InternVL2-8B` Chen et al. (2024b). Where performance metrics were available, we did not re-evaluate models on these datasets; instead, we report the results published in the original papers, official leaderboards, or model cards. For all other cases, we carefully fine-tuned and evaluated the models following the instructions provided in their respective official GitHub repositories.

**Evaluation Protocol:** All models are evaluated on the test sets of the four selected datasets. Each model receives image-question pairs, formatted within a unified prompt that includes a system message tailored to elicit the model's reasoning capabilities (Section G.1). For the Visual-TableQA dataset, we additionally construct a variant in which data is provided not as rendered images but in LaTeX code format. This textual-code version is referred to as Visual-TableQA-CIT.

For `LLaVA-Next-Llama3-8B`, `MiniCPM-V2.5-Llama3`, `InternVL2-8B`, and `Qwen2.5-VL-7B-Instruct`, we conducted two supervised fine-tuning (SFT) experiments: *(i)* using the ReachQA training split (denoted as `Model_Name + ReachQA`) and *(ii)* using the Visual-TableQA training split (denoted as `Model_Name + Visual-TableQA`). We applied Low-Rank Adapters (LoRA) Hu et al. to all linear layers, following the SFT setup and hyperparameters described in the He et al. GitHub repository when possible (Section H) in order to make a fair comparison. The fine-tuning phase for all models was limited to one epoch to ensure consistency and reduce overfitting. Exceptionally, we adopted a custom two-phase LoRA fine-tuning strategy for `Qwen2.5-VL-7B-Instruct` (see Section H), as this model was not included in the evaluation of He et al., and to better accommodate the relatively small size of our dataset.

All models are allocated a maximum of 5,000 tokens during inference to accommodate extended chain-of-thought reasoning. Model responses are evaluated using the same jury of high-performing VLMs and majority-vote protocol as described in Section 2.3. The jury confidence score, computed as the ratio of the highest vote count to the total jury size, averages above 0.87 for all models and all datasets. In addition, evaluations are run twice, to ensure reproducibility.

### 3.2 EXPERIMENTATION RESULTS

The average models accuracies are displayed in Table 2. These results reveal that:

**Visual-TableQA Effectively Evaluates Visual Reasoning Capabilities:** Model performances on Visual-TableQA follow similar trends to those observed on real-world, human-annotated datasets such as ChartQA and MATH-Vision, suggesting that synthetic datasets can effectively evaluate reasoning capabilities. A direct comparison between Visual-TableQA and its textual variant, Visual-TableQA-CIT, shows a notable performance gap: on average, models perform $+6.26\%$ better on

Figure 3: Correlation of model rankings on Visual-TableQA with those on three established datasets—ChartQA (recognition-focused), ReachQA (balanced), and MATH-Vision (reasoning-focused)—using Spearman's $\rho$ and Kendall's $\tau$ metrics. Higher values indicate stronger alignment in model performance trends. Visual-TableQA shows strong correlation with ReachQA, suggesting it effectively balances both visual recognition and reasoning, while its weaker correlation with ChartQA and MATH-Vision highlights its unique position as a comprehensive visual reasoning benchmark.

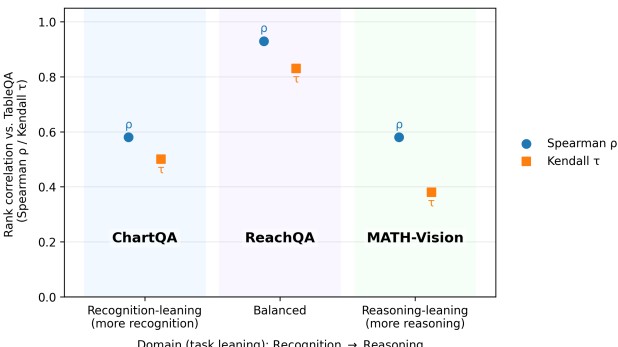

Visual-TableQA-CIT. This highlights the added challenge posed by the image-based format in Visual-TableQA, demonstrating its effectiveness at testing visual reasoning over purely textual input.

To further validate Visual-TableQA as a reasoning benchmark, we compared model rankings across datasets. For each dataset, we extracted the models (except the fine-tuned ones) performance rankings and compared them to the rankings on Visual-TableQA using two correlation measures: *(i)* Spearman's $\rho$ Lee Rodgers & Nicewander (1988): Captures monotonic consistency in rankings (regardless of exact scores); *(ii)* Kendall's $\tau$ Kendall (1948): Measures the fraction of concordant vs. discordant ranking pairs and is more robust to ties. Both metrics range from $-1$ to $1$, with values closer to $1$ indicating strong alignment in model rankings. To ensure fairness, we adjusted all scores computed with Relaxed Accuracy by subtracting 5%, before comparison. The results are shown in Figure 3.

Each dataset varies in how much it emphasizes visual recognition versus reasoning: *(i)* ChartQA $\rightarrow$ Recognition-heavy, *(ii)* ReachQA $\rightarrow$ Balanced, *(iii)* MATH-Vision $\rightarrow$ Reasoning-heavy

Interestingly, Visual-TableQA rankings align most closely with ReachQA, but not with ChartQA or MATH-Vision individually. This suggests that Visual-TableQA does not favor models that excel solely at recognition or solely at reasoning. Instead, it rewards models capable of both—making it a comprehensive benchmark for evaluating all aspects of visual reasoning.

**Visual-TableQA Effectively Transfers to Other Benchmarks:** To assess the transferability of Visual-TableQA, we investigated how fine-tuning on Visual-TableQA impacts performance across other benchmarks. As shown in Table 2, supervision from Visual-TableQA led to significant generalization beyond its native domain. Notably, it improved the accuracy of `Qwen2.5-VL-7B-Instruct` on *ReachQA* from 49.23% to 60.95%, and on *MATH-Vision* from 25.10% to 49.77%, despite these datasets not being explicitly table-focused. This finding is further supported by Table 4, which reports similar gains in generalization across three additional models: `LLaVA-Next-Llama3-8B`, `MiniCPM-V2.5-Llama3`, and `InternVL2-8B`.

However, this transferability is not reciprocal. Fine-tuning `Qwen2.5-VL-7B-Instruct` on *ReachQA* alone yields only modest in-domain gains (49.23% $\rightarrow$ 55.75%) and leads to reduced performance on both *ChartQA* and *Visual-TableQA*. This suggests that Visual-TableQA provides a more generalizable reasoning signal—rooted in layout understanding, symbolic interpretation, and multi-step reasoning—compared to standard benchmarks.

**Proprietary Models Outperform Open-Source Models on Average:** `Claude 3.5 Sonnet` achieves the highest performance across nearly all benchmarks. However, fine-tuning on Visual-TableQA substantially narrows the gap between proprietary and open-source models. No-

tably, the performance of `Qwen2.5-VL-7B-Instruct` increases significantly across all evaluated benchmarks—surpassing several state-of-the-art proprietary models, including `GPT-4o`, `GPT-4o-mini,` and `Gemini 2.5 Pro`.

# 4 DISCUSSION

## 4.1 VISUAL-TABLEQA VS REACHQA

Table 4: Performance of fine-tuned models on the two splits of the ReachQA test set: *Recognition* (Reco) and *Reasoning* (Reas), each consisting of exactly 1,000 samples. Best performances per model category are in **bold**. The values in blue are from our own evaluation using the LLM jury, while the remaining values are taken from He et al..

| Model | Reco | Reas | Model | Reco | Reas |
|---|---|---|---|---|---|
| LLaVA-Next-Llama3-8B | 17.9 | 6.5 | InternVL2-8B | 33.7 | 16.2 |
| + ReachQA | **29.6** | 11.1 | + ReachQA | **49.8** | 21.3 |
| + Visual-TableQA | 28.4 | **20.2** | + Visual-TableQA | 45.6 | **34.5** |
| MiniCPM-V2.5-Llama3 | 25.3 | 10.3 | Qwen2.5-VL-7B-Instruct | 66.20 | 33.10 |
| + ReachQA | 35.10 | 11 | + ReachQA | 69.6 | 40.30 |
| + Visual-TableQA | **36.20** | **31.50** | + Visual-TableQA | **70.3** | **50.6** |
| **Average gains** | | | | | |
| + ReachQA | +10.25 | +4.4 | | | |
| + Visual-TableQA | +9.35 | +17.68 | | | |

The ReachQA dataset is divided into two equally sized subsets: *Recognition*, which tests a model's ability to extract relevant information from charts, and *Reasoning*, which evaluates a model's capacity to understand complex and abstract data structures. Table 4 reports the performance gains of multiple fine-tuned models on these two tasks.

On average, models fine-tuned on ReachQA exhibit an accuracy improvement of $+10.25$ points on the *Recognition* task and $+4.4$ points on the *Reasoning* task. In comparison, models fine-tuned on Visual-TableQA show an average gain of $+9.35$ on *Recognition*—a comparable result—but a significantly larger gain of $+17.68$ on *Reasoning*.

This stark contrast in reasoning performance can be attributed to the presence of high-quality rationales in Visual-TableQA annotations, along with the inclusion of more complex and diverse visual structures. In other words, despite being roughly three times smaller than ReachQA in terms of sample count, Visual-TableQA places a stronger emphasis on qualitative richness over quantity. As a result, it appears to enable more effective knowledge distillation, particularly for tasks requiring symbolic interpretation and multi-step reasoning.

## 4.2 VISUAL-TABLEQA'S ADVANTAGES COMPARED TO OTHER DATASETS

Table 1 shows that only a few table-focused QA datasets—namely TAT-DQA, Table-VQA, and TableVQA-Bench—represent tables as rendered images. **Visual-TableQA** surpasses these by offering richer layout diversity, broader topic coverage, systematic visual complexity, and high-quality rationales. These attributes make it particularly effective for training models with transferable reasoning skills. Supporting this, models fine-tuned solely on **Visual-TableQA**—such as `LLaVA-Next-Llama3-8B`—demonstrated significant gains on external benchmarks (Table-VQA and TableVQA-Bench), as seen in Table 5.

Interestingly, `Qwen2.5-VL-7B-Instruct` did not follow the same performance trend: it showed degradation on tasks such as *VTabFact* (Yes/No fact verification), *VWTQ* (Wikipedia table retrieval), and *VWTQ-Syn* (synthetic variants). To understand this, we manually analyzed its errors before and after fine-tuning on *VTabFact*, categorizing them into eight types: *partial data extraction, hallucination, incoherence, misunderstanding, reasoning errors, evaluation mistakes, dataset ambiguity*, and *annotation flaws*. Results (Figure 14) show that while the total number of errors slightly increased post-finetuning, most now fall into the *incoherence* class, with all other error types significantly reduced. This suggests a sharpening of reasoning patterns but also highlights a need for

future work targeting specific error types through synthetic supervision. Further details are provided in Section I.

Beyond transferability and diversity, a key advantage of Visual-TableQA lies in its modularity and scalability as explained in Section 4.3 .

Table 5: Performance of fine-tuned models on Table-VQA test set and the four splits of the TableVQA-Bench dataset: *FinTabNetQA* (finance-related tables), *VTabFact* (table-based fact verification with Yes/No questions), *VWTQ* (information retrieval from Wikipedia tables), and *VWTQ-Syn* (synthetic visual variants of VWTQ). Best performances per model variants are shown in **bold**. Values in blue are from our own evaluation using the `DeepSeek-Prover-v2`, while remaining values are reported from Fu et al. (2025).

| Model | TableVQA-Bench | | | | Table-VQA |
|---|---|---|---|---|---|
| | FinTabNetQA | VTabFact | VWTQ | VWTQ-Syn | |
| GPT-4o | 96.8 | 78.0 | 72.8 | 82.4 | – |
| LLaVA-Next-34B | – | 71.2 | 36.4 | 38.0 | – |
| LLaVA-Next-Llama3-8B | 52.4 | 37.2 | 21.5 | 24.8 | 25.84 |
| **+ Visual-TableQA** | **56.8** | **52.0** | **33.2** | **33.6** | **28.89** |
| Qwen2.5-VL-7B-Instruct | 96.4 | **82.0** | **68.53** | **74.0** | **79.03** |
| **+ Visual-TableQA** | **97.2** | 70.6 | 61.5 | 69.6 | 75.23 |

## 4.3 SCALABILITY OF THE PIPELINE AND ITS BENEFITS FOR KNOWLEDGE DISTILLATION

This modular pipeline supports scalable generation with a clean separation of concerns—table structure synthesis, QA creation, and validation—making each component independently reusable and upgradable. By automating the entire process from table generation to jury-based quality control, Visual-TableQA provides a cost-efficient and high-quality benchmark for advancing multimodal reasoning over complex visual inputs. A central component of our pipeline is the mechanism of **cross-model inspiration** 2.2, a collaborative prompting strategy. In this process, stronger models generate layout "seeds" that guide weaker models in synthesizing structurally diverse tables, fostering novel visual configurations through iterative transfer. The same principle extends to question–answer generation: models are prompted with both layout and topical cues—often proposed by stronger models—to create new QA pairs. This enables weaker models to contribute meaningfully to the dataset by expanding the range of questions and reasoning patterns. Through this dual-inspiration process, the pipeline cultivates a collaborative multi-model co-creation space, where models of varying capabilities distill collective knowledge not through imitation, but through generative inspiration, while maintaining data quality. In this regard, **Visual-TableQA** distinguishes itself from other synthetic datasets Aboutalebi et al. (2024); Wang et al. (2024a); Li et al. (2025); He et al..

## 5 CONCLUSION

In this work, we introduced Visual-TableQA, a large-scale, open-domain, multimodal dataset designed to rigorously evaluate visual reasoning capabilities over complex table images. Building on the principles of Code-as-Intermediary Translation (CIT), we developed a fully automated, modular pipeline for generating LaTeX-rendered tables, reasoning-intensive question–answer pairs, and high-quality rationales—all verified by a jury of strong LLMs. Despite being cost-efficient (generated for under $100), Visual-TableQA offers unprecedented diversity in table structures, visual features, and reasoning depth. We showed that Visual-TableQA not only challenges existing visual language models (VLMs) but also serves as an effective training signal for improving reasoning performance. Fine-tuning on Visual-TableQA led to substantial gains across multiple benchmarks—both table-centric and general-purpose—including ReachQA and MATH-Vision, demonstrating the dataset's capacity to bridge the performance gap between open-source and proprietary models.

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
