APPENDIX

## A    EXTENDED RELATED WORKS

The vast majority of table-based QA datasets—such as HybridQA Chen et al. (2020b), WikiTable-Questions Pasupat & Liang (2015), WikiSQL Zhong et al. (2017), and AIT-QA Katsis et al. (2022)—represent tables in textual format rather than as rendered images, thereby bypassing the challenges associated with visual layout interpretation. In contrast, our work focuses exclusively on multimodal datasets—those that contain both textual and visual (image-based) information. These can generally be grouped into two main categories: real-world datasets, collected from authentic documents, and synthetic datasets, generated using automated tools. Real-world multimodal QA datasets that emphasize tables—such as TAT-DQA Zhu et al. (2022), and TableVQA-Bench Kim et al. (2024)—tend to be highly domain-specific, limiting diversity in both table layouts and question types. For example, TAT-DQA Zhu et al. (2022) combines tabular and textual data from financial reports and, while it introduces hybrid contexts for realistic reasoning, its questions rely heavily on reading textual input rather than interpreting visual structure. Similarly, TableVQA-Bench Kim et al. (2024) consists of 83% real-world tables (1,250 out of 1,500), primarily sourced from task-specific datasets such as WikiTableQuestions (information retrieval), TabFact Chen et al. (2020a) (fact verification), and FinTabNet Zheng et al. (2021) (financial data extraction). Due to its relatively small size and the specialized nature of its subsets, the dataset exhibits limited visual diversity. This limitation also extends to the remaining 16% synthetic tables, whose visual variation is restricted to basic formatting attributes such as background color, border size, font size, and style. More recently, ChartQA Masry et al. and DocVQA Mathew et al. (2020) have introduced large open-domain datasets for visual question answering. ChartQA focuses on reasoning over charts and plots; however, its tasks primarily involve shallow reasoning and do not reflect the structural complexity and layout diversity found in real-world tables. In contrast, DocVQA offers greater diversity in document layouts and structures, but lacks significant visual challenge—recent VLMs, including relatively lightweight models like `Qwen2.5-VL-7B-Instruct` Team (2025), achieve over 94% accuracy on this benchmark.

Among synthetic datasets, *MultiModalQA* Talmor et al. (2021) stands out as the only open-domain resource focused on tables. It combines real-world figures, diagrams, and text passages sourced from Wikipedia, with QA pairs crafted to assess both reasoning and visual comprehension. Although it incorporates real content, the dataset is considered synthetic due to the way it links independent modalities and generates QA pairs through formalized templates. However, this approach results in limited diversity, as the questions are derived from a finite set of templates. In contrast, *Table-VQA* Tom Agonnoude (2024) is a fully synthetic dataset generated using state-of-the-art LLMs. Nonetheless, it lacks visual diversity and complexity—its tables follow similar formats and are predominantly centered around technical domains such as statistics, physics, and algorithms, all of which are heavily numerical in nature.

Most of real-world datasets rely heavily on manual labeling, data collection, and preprocessing—factors that significantly constrain their scalability. Recently, ReachQA He et al. introduced a more scalable and innovative approach through its Code-as-Intermediary Translation (CIT) pipeline. This method generates synthetic charts and reasoning questions by leveraging textual intermediaries such as Python code, demonstrating that advanced reasoning capabilities of large language models (LLMs) can be effectively transferred to visual models. While ReachQA successfully addresses both scalability and reasoning complexity, its approach is tailored to chart-based visualizations and does not extend to structured tabular data.

To summarize, existing table-based benchmarks consistently fall short in one or more key areas: visual diversity, reasoning depth, or scalability. Notably, aside from MultiModalQA, there is no open-domain dataset designed to evaluate model performance on rendered table images, despite their prevalence in real-world settings such as reports, academic papers, and spreadsheets. In this work, we introduce **Visual-TableQA**, a multimodal open-domain synthetic dataset specifically created to assess reasoning capabilities over table images using LLMs

## B    CONSIDERATIONS AND LIMITATIONS

The main limitations of this work relate to the use of Code-as-Intermediary Translation (CIT) He et al. and the assessment of data quality. While we adopted LaTeX as an intermediate representation for tables, its expressiveness is limited when handling more complex or visually rich images. Developing a robust, bidirectional image-to-text encoding system remains an open and promising area for future research. In terms of data quality evaluation, although automatic metrics such as ROSCOE provide useful insights, they are not yet as reliable as human judgment. As a result, human annotators continue to play a critical role in ensuring high-quality data, especially when scaling synthetic datasets for reasoning tasks.

In addition, we also observed that certain models, such as `Qwen2.5-VL-7B-Instruct`, did not consistently benefit from Visual-TableQA supervision across all downstream tasks, highlighting a potential limitation in generalization that warrants further investigation.

## C    TABLEQA LAYOUT AND TOPIC DIVERSITY

As described in Section 2.2, we sampled 5,000 distinct topics using `GPT-4o` to serve as inspirations for table generation. To better illustrate topic diversity, we grouped these topics into 20 semantic clusters using the K-Means algorithm. Figure 5 displays a 2D projection of these clusters, where each color represents a distinct semantic group. For the 12 largest clusters, we highlight representative topics to give a sense of their thematic content.

In addition, Figure 4 shows the cumulative percentage of topics covered as clusters are added in descending order of size. The smooth progression of the curve indicates that the clusters are relatively uniform in size, confirming a balanced distribution of topic diversity throughout the dataset.

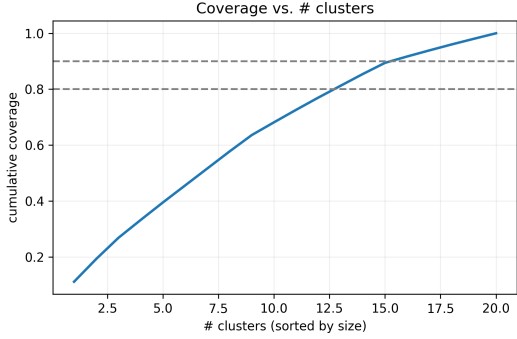

Figure 4: Cumulative topic coverage as clusters are added by descending size. The uniform slope indicates an even distribution of topics across clusters.

To illustrate the diversity of table layouts produced by our pipeline, Figure 6 displays a side-by-side comparison between the initial seed tables used in the first generation iteration and a sample of layouts generated in subsequent steps. The wide range of structures highlights the pipeline's capacity to create rich and varied visual designs from limited starting templates.

## D    TABLE GENERATION SETTINGS

The first stage of the generation pipeline involves **LLM-1**, which is responsible for producing new tables based on given inspirations. Specifically, it receives one LaTeX-formatted table as a *layout inspiration* and three distinct *topic inspirations*. Based on these, it generates three new tables, each aligned with one of the provided topics while drawing structural influence from the layout example. The full prompt used to guide LLM-1 during this step is shown in Figure 7.

The second stage of the generation pipeline involves **LLM-2**, which is responsible for producing question–answer (QA) pairs based on a single LaTeX table. The full prompt used to guide LLM-2 is shown in Figure 8.

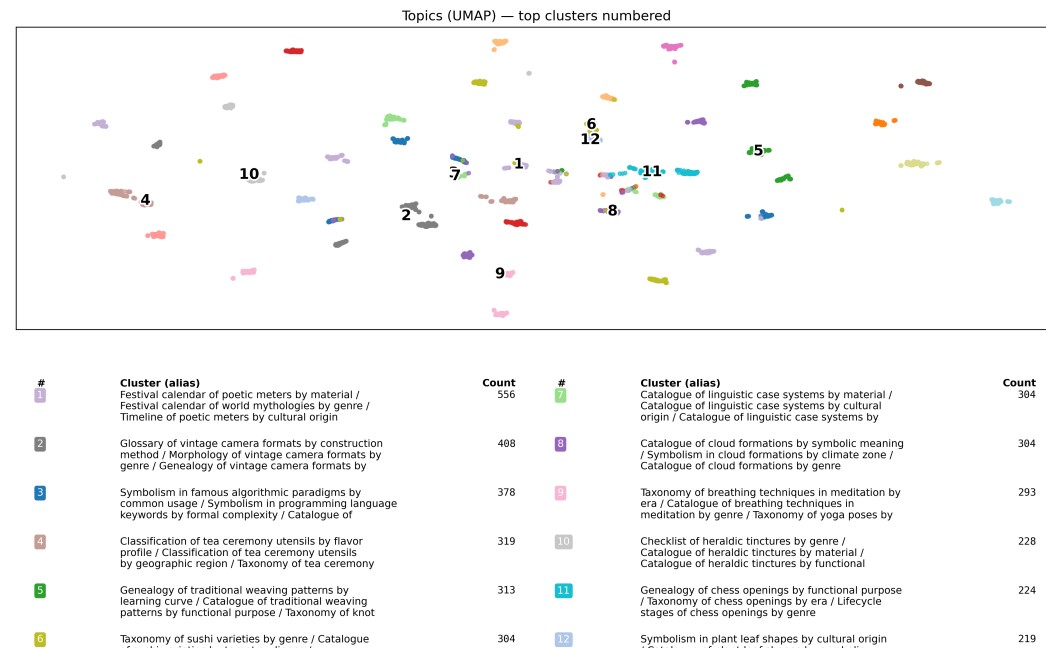

Figure 5: 2D projection of the 5,000 topics using UMAP and K-Means clustering. Each color denotes a semantic cluster. Representative topics are listed for the 12 largest clusters to illustrate diversity.

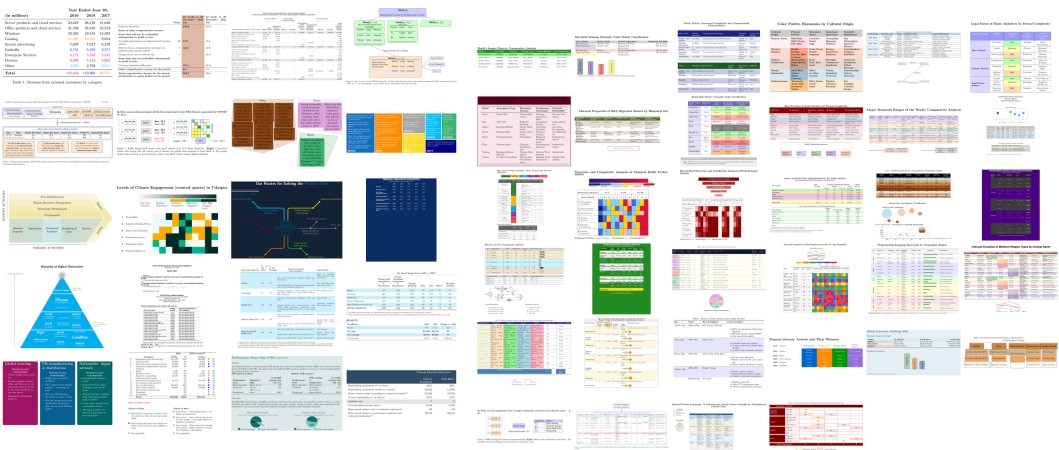

(a) Layout seeds used for the initial table generation.

(b) Sample of diverse layouts generated by our pipeline.

Figure 6: Visual diversity of table layouts. Left: seed layouts used during the first iteration of table generation. Right: layouts generated through cross-model inspiration and iterative refinement.

To encourage creativity while maintaining factual accuracy, the temperature parameter for each model during this phase is set to 0.7.

In Table 6, we present the average validity rates of the generated QA pairs for each model involved in the QA generation phase.

```
You are an expert in generating synthetic datasets composed of LaTeX-formatted tables,
    optionally accompanied by illustrative diagrams. Your task is to produce structured
    content suitable for data-centric documents, ensuring each table (and diagram, if
    included) is clear, well-organized, and visually informative.
Your final output should start with ```json and end with ``` as plain text, not just formatting
    . Like this:

```json
{
  "table_1": "BEGIN_LATEX
<LaTeX code for table 1 (with/without diagram) here>
END_LATEX",
  "table_2": "BEGIN_LATEX
<LaTeX code for table 2 (with/without diagram) here>
END_LATEX",
  "table_3": "BEGIN_LATEX
<LaTeX code for table 3 (with/without diagram) here>
END_LATEX"
}
```

Requirements:
    The tables and diagrams will be used to generate reasoning questions. Therefore:

        - If topic inspirations are supplied, ensure every generated table aligns with those
            topics.
        - Each LaTeX output must primarily consist of a table. Include a diagram only if it
            meaningfully complements the table; avoid adding one unnecessarily. Do not generate
            diagrams alone. If a diagram is empty or non necessary DON'T INCLUDE it.
        - Keep any diagram minimal---smaller than the table, chart-free, and purely illustrative
            ---serving only to reinforce the table's content without adding new information.
        - Each table and their diagram must contain realistic, domain-relevant content. They
            must be self-contained, include a clear descriptive title and not rely on external
            data to compile.
        - The type of information presented should be diverse---such as numerical data or
            qualitative. The variety and richness of visual elements is essential to the
            overall quality of the table and their diagram. Table quality should also come with
            a large number of rows and columns.
        - Table and diagram layouts should be creatively designed---taking inspiration from
            reference example (when provided) but incorporating meaningful variations such as
            colors, multi-row or multi-column cells, custom formatting adjustments, or any
            other visual enhancement that promotes structural diversity.
        - Table layouts should be at least as complex as the example provided, don't try to
            simplify (diagrams are not mandatory). Table complexity should also come with a
            large number of rows and columns.
        - Do NOT escape any characters in the LaTeX code. The LaTeX must be written as plain
            text, exactly as it would appear in a .tex file, with real line breaks and single
            backslashes (\), not JSON-escaped.
        - All LaTeX tables and diagrams must be constrained to fit entirely within the printable
            area of a standard A4 page when compiled to PDF, without overflowing horizontally
            or vertically. Use appropriate formatting techniques such as adjusting column
            widths, reducing font size, or enabling landscape mode if necessary but NEVER
            rotation.
        - Make sure each LaTeX table and diagram includes all required \usepackage declarations
            and is enclosed within a complete, compilable LaTeX document structure, including
            the appropriate preamble and \begin{document}...\end{document} block.
        - Make sure each LaTeX codes start and end with BEGIN_LATEX and END_LATEX, respectively.
        - Make sure to wrapp your final answer with ```json at the beginning and ``` at the end.
```

Figure 7: LLM prompt used for table generation.

## E   COMMON ANOMALIES IN GENERATED TABLES

As summarized in Table 3, several tables generated by different LLMs required post-processing adjustments to ensure visual clarity, LaTeX correctness, and topic alignment. Below, we categorize the most common types of anomalies observed during generation:

### LAYOUT AND FORMATTING ISSUES

- **Table overflow:** Tables exceeding page margins due to improper column widths or missing column dimensioning.

```
You are an expert in generating question--answer pairs from LaTeX-formatted. Your task is to
    create a structured dataset consisting of visually challenging, reasoning-based questions
     and their corresponding answers derived from a given LaTeX formatted table with optional
     diagram.

Input:

You will be provided with a sample LaTeX table as context. Based on this table or diagram,
    your goal is to generate a JSON object with the following structure:

  questions: A python list of 3 challenging questions that require reasoning and analysis
      based ONLY on the data presented in the table and the optional diagram. The questions
      must be answerable using ONLY the information in the table or diagram(no extra
      knowledge).
  answers: A python list of 3 detailed answers to the 3 questions, including a clear chain of
      thought explaining the reasoning process.

Requirements:

  All questions must be relevant to the table's context and designed to test deeper
      understanding or inference.
  When possible, all questions should make full use of the visual or structural elements of
      the table or diagram (such as rows, columns, headers, colors, patterns, diagrams etc.)
       while maintaining clear relevance to the table's content.
  Questions must be clear and answerable with an objective methodology, no subjective
      question.
  All entries (both questions and answers) should be returned as lists of string values.
  The global result should be a single JSON object wrapped in a markdown code block using ```
      json at the beginning and ``` at the end, and containing all two key-value pairs.
  This means your output should start with ```json and end with ``` as plain text, not just
      formatting.
```

Figure 8: LLM prompt used for QA generation.

Table 6: Average QA pair validity across different QA generation models. Accuracies are computed from a sample of at least 500 QA pairs per model.

| Model | Acc. (%) |
| --- | --- |
| Llama 4 Maverick 17B-128E Instruct | 88 |
| Gemini 2.0 Flash | 89.1 |
| Gemini 2.5 Flash | 93.1 |
| Gemini 2.5 Pro | 89.3 |
| GPT-4.1 | 90.4 |
| Qwen3-30B-A3B | 76.6 |
| Qwen3-QwQ-32B | 92.6 |
| DeepSeek-R1-Distill-Qwen-32B | 73.4 |
| DeepSeek-R1T-Chimera | 89.4 |
| Claude Sonnet 4 | 91 |
| Claude 3.5 Haiku | 90.2 |
| Grok 3 Beta | 89.4 |
| Reka Flash 3 | 79 |

- **Text overflow:** Cell content spilling outside the cell boundary, especially in narrow columns or with long strings.

- **Invisible content:** Multirow cells with background colors that obscure cell text (e.g., white text on white background).

- **Improper horizontal lines:** \midrule or \hline splitting across multirow cells, breaking visual coherence.

CONTENT RELEVANCE AND CORRECTNESS

- **Empty or irrelevant tables:** Tables with placeholder content or unrelated to the assigned topic.

- **Incorrect topic alignment:** Generated tables that do not match the intended topic inspiration.

- **Duplicate outputs:** All three tables generated for a prompt are identical or nearly identical in structure/content.

- **Missing external resources:** References to images or figures not included or available in the output.
- **Incorrect LaTeX syntax:** Math symbols placed outside of math environments, leading to compilation errors.

DIAGRAM-SPECIFIC ISSUES

- **Missing tables:** Some generations return only a diagram without an accompanying table.
- **Node placement errors:** Overlapping or misaligned nodes in TikZ diagrams.
- **Arrow misplacement:** Arrows that do not connect to correct nodes or that overlap diagram elements improperly.
- **Legend/title confusion:** Titles or legends positioned incorrectly or detached from the relevant diagram elements.
- **Visual inconsistencies:** General drawing flaws, such as missing anchors, inconsistent line styles, or unintended overlaps.

These issues highlight the need for a validation loop in the TableQA pipeline and justify the inclusion of human verification stages to ensure dataset quality.

# F    ROSCOE METRIC SCORES

In addition to the LLM jury validation of the Visual-TableQA dataset, we conducted a complementary quality assessment using the ROSCOE framework Golovneva et al.. This evaluation measures step-by-step reasoning coherence across multiple dimensions, including semantic alignment, logical consistency, and contextual grounding. The resulting scores, reported in Table 7, further support the reliability and high quality of the generated tables and QA pairs, reinforcing the effectiveness of our data generation pipeline.

Table 7: ROSCOE Golovneva et al. reasoning metrics averaged over the whole dataset. The "Direction" column indicates whether higher or lower values correspond to better performance for each metric.

| Metric | Direction | Mean | Std |
|---|---|---|---|
| **Semantic Adequacy ($\uparrow$)** | | | |
| Faithfulness-Step | $\uparrow$ | 0.99 | 5e-4 |
| Informativeness-Step | $\uparrow$ | 0.99 | 5e-3 |
| Informativeness-Chain | $\uparrow$ | 0.98 | 1.4e-2 |
| Faithfulness-Token | $\uparrow$ | 0.99 | 2e-3 |
| **Avg** | | **0.99** | |
| **Redundancy & Risk ($\downarrow$)** | | | |
| Repetition-Token | $\downarrow$ | 0.06 | 0.12 |
| Repetition-Step | $\downarrow$ | 0.06 | 0.12 |
| **Avg** | | **0.06** | |
| **Logical Inference ($\uparrow$)** | | | |
| Discourse-Representation | $\uparrow$ | 0.68 | 0.41 |
| Coherence-Step | $\uparrow$ | 0.7 | 0.40 |
| **Avg** | | **0.69** | |
| **Fluency & Perplexity ($\downarrow$)** | | | |
| Perplexity-Step | $\downarrow$ | 0.01 | 0.01 |
| Perplexity-Chain | $\downarrow$ | 0.05 | 0.03 |
| Perplexity-Step-Max | $\downarrow$ | 8e-3 | 8e-3 |
| **Avg** | | **0.02** | |
| **Grammaticality ($\uparrow$)** | | | |
| Grammar-Step | $\uparrow$ | 0.96 | 0.05 |
| Grammar-Step-Max | $\uparrow$ | 0.9 | 0.13 |
| **Avg** | | **0.93** | |

# G  LLM Jury Reliability

Following recent studies He et al.; Fu et al. (2025); Verga et al. (2024), we adopt a high-performing LLM jury combined with a majority-vote strategy to evaluate model predictions. This multi-model jury setup enhances evaluation robustness and mitigates individual model biases. The configuration details for all jury models are provided in Section G.1. As argued in Verga et al. (2024), aggregating judgments from several strong LLMs yields more consistent and reliable evaluations than relying on a single model. In Section G.2, we discuss the challenges of LLM-based evaluations and we report a detailed comparison between our LLM jury and human annotators on `Qwen2.5-VL-7B` predictions across multiple benchmarks. In Section G.3, we provide a detailed analysis of jury-to-jury agreement across both table and QA pair quality assessments, as well as benchmark evaluations. The results reveal strong inter-annotator alignment, validating the consistency and effectiveness of our jury-based evaluation protocol.

## G.1  LLM Jury Settings

LLM juries were involved at two key stages of the pipeline: (i) **quality assurance**, where generated tables and QA pairs were validated before inclusion in the dataset, and (ii) **evaluation benchmarking**, where model responses were assessed for accuracy and reasoning quality.

The prompt used for quality assurance is shown in Figure 9, while the prompt used for evaluation during benchmarking is shown in Figure 10. Each jury consisted of multiple high-performing vision-language or reasoning-capable LLMs, and final decisions were made via majority voting.

To ensure consistency and reproducibility across evaluations, all LLM jury calls were executed with a temperature setting between 0.0 and 0.1.

We observed a notable drop in judgment accuracy when LLM juries were instructed to return only a structured JSON verdict without any preceding explanation. In particular, `Mistral-large` systematically omitted its reasoning whenever the keyword `JSON` was included in the prompt. This issue was effectively mitigated by explicitly instructing models to provide a rationale prior to their final decision and by avoiding any direct mention of `JSON` in the prompt, except for `GPT-4.1` which remained robust under such formatting. Including explicit reasoning significantly improved the reliability and depth of model evaluations, likely by reducing shallow assessments and prompting more thoughtful judgments.

```
You are a reasoning question answer expert. You will be given a LaTeX formated table with/
    without diagram, a list of 3 topics, and a pair of a question and its answer.

Your task is to evaluate the pair of question answer based solely on the data in the LaTeX
    code and these criteria:

  1) Does the LaTeX code contain a Table (not some charts alone or diagrams alone) ?
  2) Does the table, any optional diagrams, and the rest of the document are on one single
      topic from the provided list of topics, and internally consistent (be careful to off-
      topic diagrams)?
  3) Is the question clear and related to the table or the diagram?
  4) Is the answer (including its reasoning) totally valid and does it actually respond to
      the question?
  5) Is the answer FULLY supported by and ONLY BY the table or diagram data (no extra
      knowledge)?

If the five criteria are true, mark the pair as correct.
If one of the criteria is not met, mark it as incorrect.

Think step by step and conclude with your decision and the index of the criterium not met (if
    none, index is 0) as follows:
JSON_mention
{{"decision": [0, index_of_the_criterium_not_met]}} for incorrect or {{"decision": [1, 0]}}
    for correct.
```

Figure 9: LLM prompt used for QA evaluation.

```
You are an expert evaluator of question-answer pairs. You will be given a question a model's
    answer and a ground truth answer (reference).
Evaluate the answer based on these criteria:
1) Is the model's answer logically consistent?
2) Does the model's answer convey the same meaning as the ground truth?
If the two criteria are true, mark the pair as correct. If one of the criteria is not met,
    mark it as incorrect.

Think step by step and conclude with your verdict and the index of the criterium not met (if
    none, index is 0) as follows:
{{"verdict": [0, index_of_the_criterium_not_met]}} for incorrect or {{"verdict": [1, 0]}} for
    correct.

Question: {question}
Answer: {answer}
Ground Truth: {ground_truth}
Response:
```

Figure 10: LLM prompt used for Benchmark evaluation.

### G.2   LLM JURY LIMITATIONS AND MITIGATION STRATEGIES

Our evaluation protocol measures alignment between a model's prediction and the reference answer, leveraging LLM juries for semantic comparison. This strategy is highly cost-effective, enabling scalable automatic assessments on reasoning tasks. However, it comes with important limitations. A model's response may rely on external world knowledge rather than explicitly extracting information from the table image. In such cases, LLM juries lack the ability to verify whether the response is grounded in the table itself, as they do not have access to the visual context. This weakness makes the evaluation pipeline vulnerable to false positives, especially when the model outputs a factually correct answer that is not actually derivable from the table content. This could be mitigated by including rendered table images alongside the question and model prediction at evaluation time—giving LLM juries full visual grounding. Unfortunately, While effective, this approach is significantly more costly in terms of API calls and inference latency, making it challenging to scale on large datasets. These observations underscore the need for hybrid evaluation strategies combining automatic LLM-based judgments with human verification, in order to quantify the risk of jury-related errors at scale.

To rigorously assess this risk, we conducted a manual evaluation of `Qwen2.5-VL-7B-Instruct`'s predictions on both Visual-TableQA and the *VTabFact* split of TableVQA-Bench, one of the benchmarks where the model exhibited degraded performance after fine-tuning. Our manual analysis serves as a ground truth reference to assess the reliability and limitations of LLM-based jury evaluation. Our evaluation considers not only the final answer, but also the validity of the reasoning process: even predictions with the correct answer are marked as incorrect if the chain of thought is flawed. We used the manual analysis to validate the judgments made by the LLM juries. We identified two types LLM Juries errors:

- *False Positive:* The LLM jury mistakenly accepts an incorrect model prediction as correct.
- *False Negative:* The LLM jury mistakenly rejects a correct model prediction as incorrect.

The errors distribution are detailed in Table 8. The results reveal that the LLM jury aligns with human judgment within a reasonable margin of $4.7\%$, even under our strict annotation protocol that penalizes incorrect reasoning regardless of the final answer. A more relaxed evaluation criterion would yield an even smaller discrepancy. These findings reinforce the reliability of our LLM jury setup, demonstrating its effectiveness as a scalable proxy for human evaluation.

Table 8: Misclassifications by LLM juries on Visual-TableQA and *VTabFact*. Percentages are calculated relative to the total number of evaluated examples in each dataset.

| Error Type | Visual-TableQA | *VtabFact* |
|---|---|---|
| False Positive | 3.45% | 1.6% |
| False Negative | 1.2% | 0.0% |
| Total | 4.65% | 1.6% |

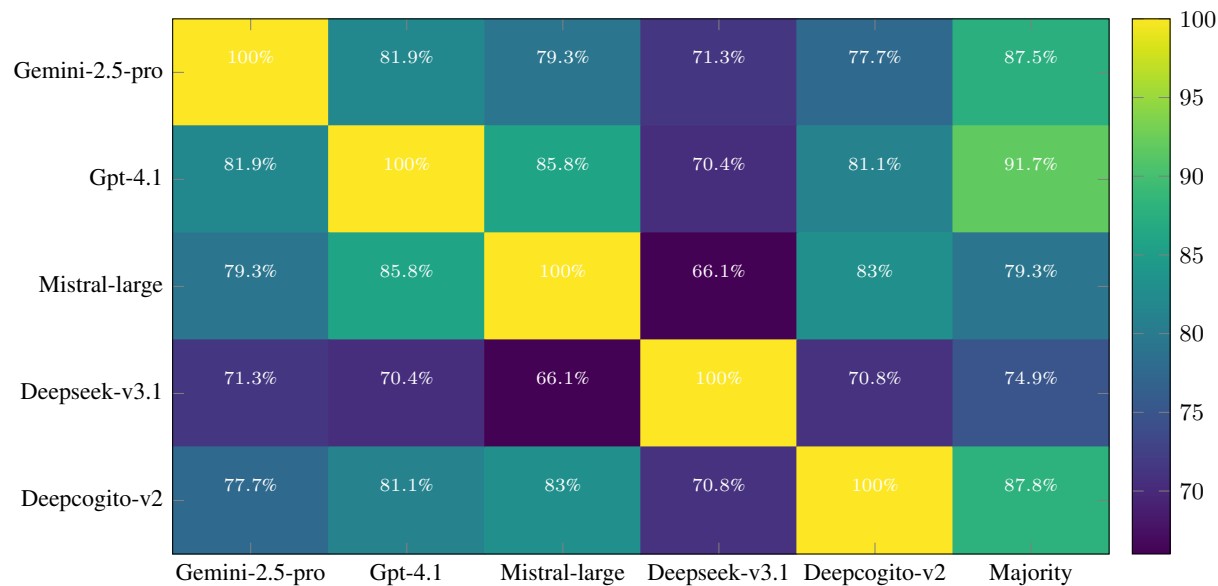

Figure 11: Pairwise agreement (%) between LLM juries, plus alignment with majority vote for tables and QAs quality assessment

### G.3 LLM Jury Agreement Analysis

Figure 11 presents a detailed breakdown of agreement levels between individual LLM juries, as well as their alignment with majority-vote annotations. This analysis was conducted across both table and QA pair quality assessments. The analysis reveals a spectrum of consistency across juries, with `GPT-4.1` emerging as the most reliable, likely due to its robust handling of edge cases. Among all models, proprietary LLMs such as `Gemini-2.5-pro` and `GPT-4.1` show the strongest alignment with the majority vote, while `Deepseek-v3.1` exhibits the weakest agreement. Notably, pairwise jury agreement patterns appear correlated with the models' reasoning capabilities. Despite variability in alignment strength, all juries demonstrate a meaningful degree of concordance with the majority, underscoring the robustness of our collective evaluation protocol.

Conversely, Figure 12 shows consistently strong jury agreement across all models for benchmark evaluations, with no notable divergence between proprietary and open-source LLMs. This can be attributed to the relatively simpler nature of the task (semantic comparison between model predictions and ground truth) compared to the more complex reasoning required for evaluating table-question-answer triples, as analyzed in Figure 11.

### G.4 Evaluator Consistency Compared to the Literature

In line with recent studies He et al.; Fu et al. (2025); Verga et al. (2024); Agarwal & Ciucă (2025), we employed a high-performing LLM jury with a majority-vote strategy to evaluate model predictions. The jury consisted of `Mistral-large`, `Deepseek-v3.1`, `Gemini-2.5-pro`, `GPT-4.1`, and `Deepcogito-v2`. In Table 2, baseline values (in black) are reported from He et al., who used `GPT-4o` as the sole evaluator. While our evaluation pipeline involves a broader and more powerful set of models, making it arguably more reliable and robust (Verga et al., 2024), we still consider the two evaluation protocols broadly comparable. In fact, due to the stricter majority-vote requirement across diverse models, our approach may even yield more demanding or rigorous evaluations. This comparability also holds for Table 5, where baseline performances (in black) are taken from Fu et al. (2025), who employed `GPT-4` for their evaluation. We argue that despite methodological differences, all evaluations are consistent enough to be analyzed jointly for comparative purposes.

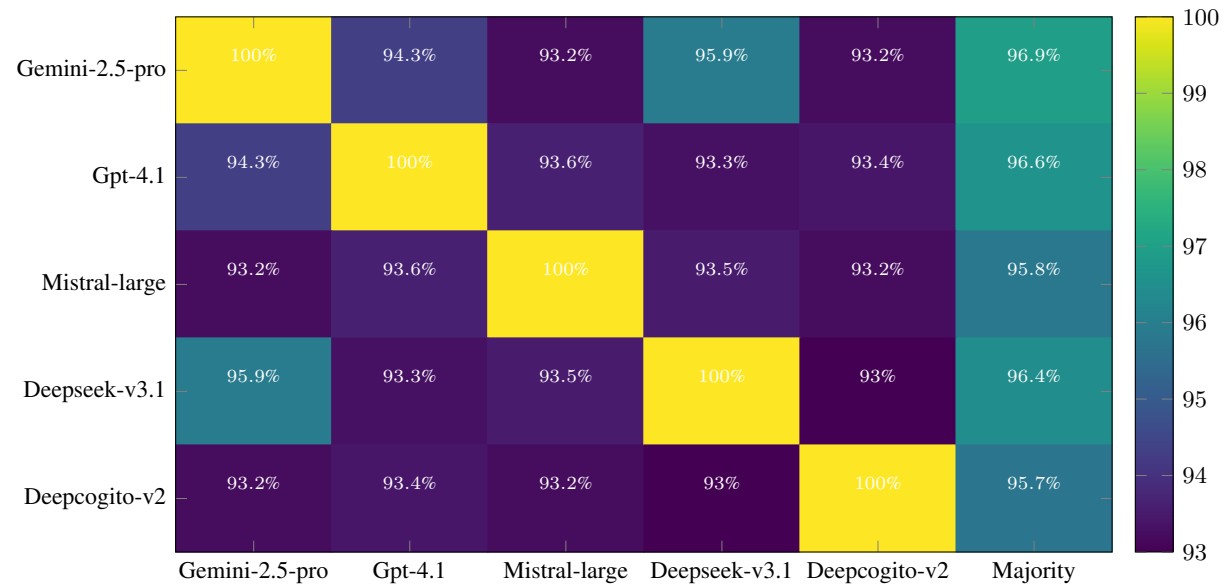

Figure 12: Pairwise agreement (%) between LLM juries, plus alignment with majority vote for benchmark evaluation

## H  MODEL FINETUNING HYPERPARAMETERS

The hyperparameters used for LoRA are reported in Table 9. For `Qwen2.5-VL-7B-Instruct`, we employed a two-phase fine-tuning strategy. In the first phase (Tier A), we adapted the text-side modules together with the multi-modal projector, while keeping the vision tower frozen. In the second phase (Tier B), we further enabled LoRA on the attention projections of the last four vision blocks, leaving the remaining vision layers untouched.

Table 9: Hyperparameters Used for Fine-Tuning with LoRA. More details in our GitHub repository. Abbreviations: lr=learning rate, r= LoRA rank, $\alpha$= LoRA $\alpha$, Targets=targets modules for LoRA.

| Model | lr | r | $\alpha$ | Targets |
|---|---|---|---|---|
| LLaVA-Next-Llama3-8B | | 16 | 8 | all-linear |
| MiniCPM-V2.5-Llama3 | 2e-5 | | | (llm frozen) |
| InternVL2-8B | | 16* | 32* | |
| Qwen2.5-VL-7B-Instruct | 1e-4 | 16 | 8 | Tier A: projector modules** |
| | 2e-5 | 8 | 32 | Tier B: last 4 vision blocks' attention |

\* The InternVL training source code sets the LoRA alpha as twice the LoRA rank, as shown in their official implementation here. We followed this convention for full reproducibility and assumed that other baselines applied the same rule.
\*\* We left the Vision Tower untouched as it significantly degraded model performance.

## I  ERROR TAXONOMY OF MODEL PREDICTIONS

To better understand the failure modes of `Qwen2.5-VL-7B-Instruct`, we conducted a fine-grained manual analysis of its predictions, both before and after fine-tuning on Visual-TableQA and the *VTabFact* split of TableVQA-Bench. This section is organized as follows: *(i)* we define the full set of error types used in our annotation protocol (Section I.1); *(ii)* we present a comparative analysis of errors observed on *VTabFact* (Section I.2).

## I.1 ERROR CATEGORIES

We classified the observed errors into eight categories (Figure 13):

- **Partial Data Extraction:** The model overlooks some relevant entries (e.g., stops counting too early, Figure 13a).
- **Hallucination:** The model references information not present in the table (Figure 13b).
- **Incoherence:** The model extracts the correct data but then misinterprets it or contradicts itself later (deductive error, Figure 13c).
- **Misunderstanding:** The model produces factual statements that do not actually answer the question (Figure 13d).
- **Faulty Methodology/Reasoning:** The reasoning is too shallow, or the model fails to satisfy all the constraints of the query (Figure 13e).
- **Ambiguity (Gray Area):** Both the ground truth and the model's prediction can be reasonably justified (Figure 13f).
- **Dataset Mistake:** The original dataset contains annotation or label errors.

## I.2 ERROR ANALYSIS ON VTABFACT

The results of our analysis are shown in Figure 14. The side-by-side comparison of Figure 14a and Figure 14b reveals a significant shift in error distribution after fine-tuning: although the total number of errors increases from 67 to 88 (out of 250 samples), most newly introduced errors concentrate in the *Incoherence* and *Hallucination* categories, while all other error types show a marked decline. In particular, *Faulty Methodology/Reasoning* errors are significantly reduced, indicating that the fine-tuned model exhibits more consistent and structured reasoning patterns. This trend is further supported by Figure 14c, which shows the distribution of errors corrected by the fine-tuned model: improvements span across all error categories. In contrast, Figure 14d shows that newly introduced errors after fine-tuning are largely concentrated in only a 2 categories. We attribute the sharp rise in *Incoherence* errors to two key factors:

**Degraded Arithmetic Capability:** A recurrent issue post-finetuning involves numerical comparison errors or basic math computation errors, leading to faulty deductions despite otherwise correct reasoning steps. Such errors were less frequent in the pretrained model.

**Answer Template Behavior:** The fine-tuned model tends to state its final answer before providing its chain of thought, often declaring an incorrect answer followed by a valid justification that leads to the correct conclusion. Despite the correctness of the rationale, we count such instances as errors. In contrast, the pretrained model typically presents its reasoning first, then concludes—leading to fewer coherence violations.

## J VISUAL-TABLEQA SAMPLE

Table 10 gives some more detailed examples of our dataset samples.

## K IMAGE-TO-LATEX DATASET

We have also released an additional dataset, **Img2TeX**, available on Hugging Face. It contains all the table images generated during the construction of **Visual-TableQA**, along with their corresponding LaTeX source code. This dataset is intended to complement the work of Kale & Nadadur (2025), which focuses on evaluating models' ability to generate LaTeX from textual prompts. In contrast, **Img2TeX** targets the task of generating LaTeX documents from visual (image-based) inputs. This distinction opens up a new evaluation pathway for vision-language models (VLMs), particularly those aiming to learn structured document generation from visual cues. As such, **Img2TeX** serves as a valuable benchmark for assessing the visual-to-structured-text generation capabilities of multi-modal models.

**Q:** there be 6 artist with claimed sale of 150 million?

**A:** To determine if there are 6 artists with claimed sales of 150 million, let's count the entries in the "claimed sales" column where the value is 150 million:
1. Eagles [...] 6. Aerosmith
There are indeed 6 artists listed with claimed sales of 150 million.

(a) Partial Data Extraction

**Q:** datia 's constituency number be lower than gohad 's by 10

**A:** To determine if Datia's constituency number is lower than Gohad's by 10, we need to compare their respective constituency numbers.
1. Identify the constituency number for Datia: The table shows that Datia's constituency number is 20. [...]

(b) Hallucination

**Q:** the majority of the district be new creation and do not have an incumbent

**A:** To determine if the majority of districts were newly created and did not have an incumbent, let's analyze the "incumbent" column:
1. **New York 1**: Incumbent: John Smith [...] 6. **New York 12**: Incumbent: David Thomas [...] From this analysis: - There is only one district (New York 1) that has an incumbent. [...]

(c) Incoherence

**Q:** of the 18 game play , 8 result in 1 or more team be shut out

**A:** To determine how many of the 18 games resulted in at least one team being shut out, we need to analyze each game's score:
1. Birmingham vs Port Vale: 2-0 (Birmingham shut out) 2. Bury vs Exeter City: 1-2 (Exeter City shut out) [...]

(d) Misunderstanding

**Q:** the smallest crowd the charger play in front of be 44826, a road game against the raider

**A:** To determine if the smallest crowd the Chargers played in front of was 44,826 during a road game against the Raiders, let's follow these steps:
1. Identify the games where the Chargers played against the Raiders: [...] 2. Check the attendance for each game: [...] 3. Compare the attendance figures: [...]

(e) Faulty Methodology

**Q:** hd 178428 's arrival date be 10 year later than that of hd 190406

**A:** [...]
Since the difference is 10 years and 10 months, it is not exactly 10 years. Therefore, HD 178428's arrival date is not 10 years later than that of HD 190406.
**Ground Truth:** Yes

(f) Gray Area

Figure 13: Illustration of the Error Categories

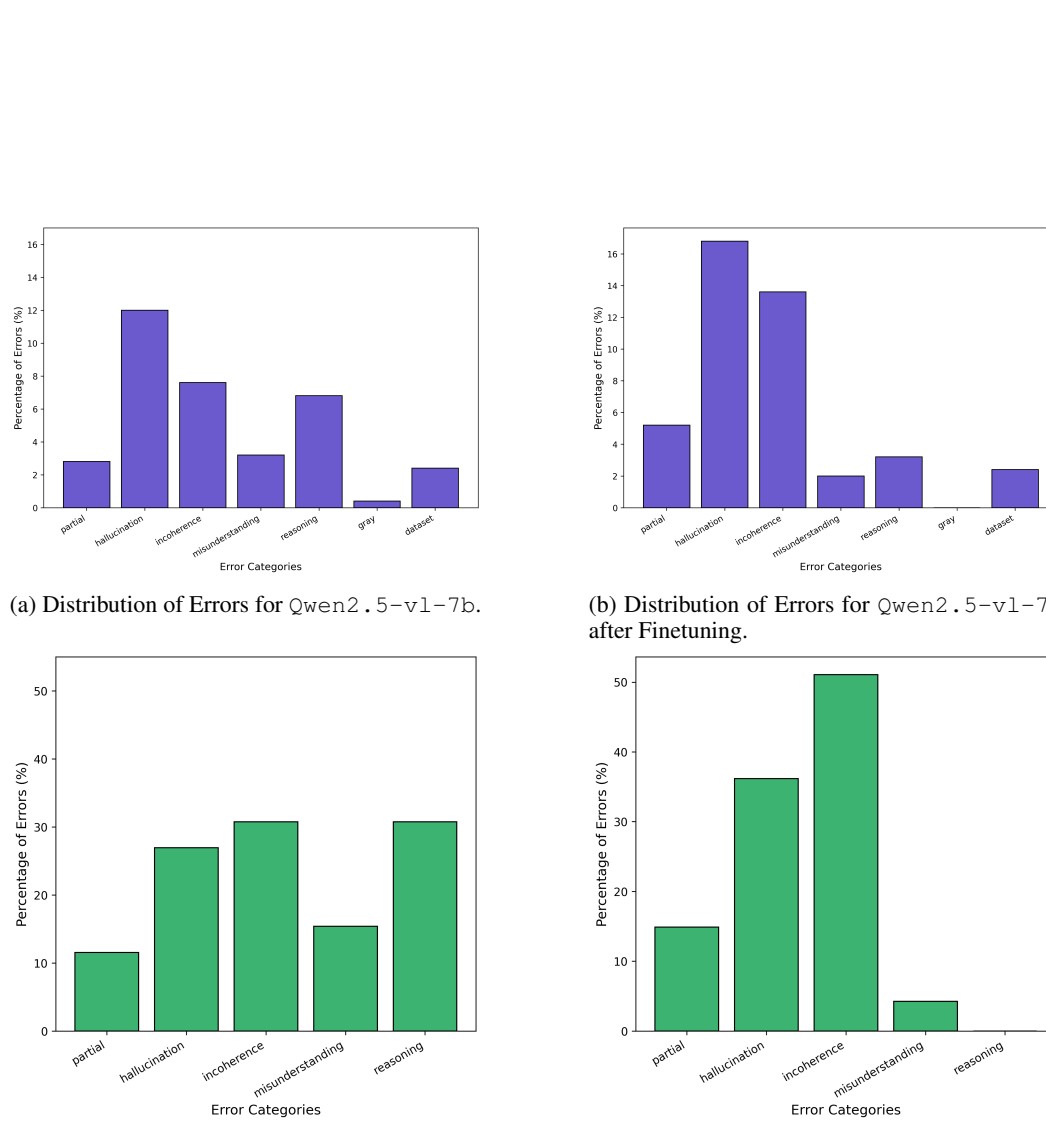

(a) Distribution of Errors for `Qwen2.5-vl-7b`.

(b) Distribution of Errors for `Qwen2.5-vl-7b` after Finetuning.

(c) Distribution of Errors corrected by Finetuning.

(d) Distribution of Errors introduced after Finetuning.

Figure 14: Comparison of error distributions between the pretrained `Qwen2.5-VL-7B` model (left) and its finetuned version (right). Percentages in the first row refer to the full *VTabFact* set. Percentages in the second row correspond to: (c) samples where the finetuned model was correct but the pretrained model was not, and (d) samples where the pretrained model was correct but the finetuned model was not.

Table 10: Sample of reasoning-intensive QA pairs. The first row's question and answer are truncated for readability. These questions address multiple visual aspects and extend beyond simple information extraction to test interpretive reasoning, as illustrated in the second row with a "how" question.

| Table | Question | Answer |
|---|---|---|
| Table 1: Comprehensive Guide to Coffee Brewing Techniques Across Skill Levels (Brewing Method, Difficulty, Equipment Required, Key Technique, Time, Visual) | Based on the visual patterns and row coloring in the table, what relationship exists between the alternating yellow-highlighted rows and the brewing methods' characteristics [...]? | Looking at the table's visual structure, the yellow-highlighted rows (rows 2, 4, 6, and 8) correspond to French Press, Chemex, Espresso, and Cold Brew methods respectively. These methods represent distinct extraction categories: French Press uses immersion, Chemex uses slow filtration, Espresso uses high-pressure extraction, and Cold Brew uses cold water extraction. The alternating color pattern appears to highlight methods that represent fundamental brewing principles or landmark techniques within each difficulty tier.[...] |
| 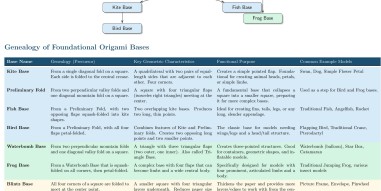 Genealogy of Foundational Origami Bases | How does the genealogy of origami bases demonstrate a progressive complexity of folding techniques, and what key transformational steps are evident in the progression from a square sheet to more complex bases? | The genealogy reveals a systematic progression from simple to complex bases, starting with the fundamental Square Sheet. The Preliminary Fold introduces diagonal and perpendicular creases, serving as a critical intermediate step. This base enables more complex transformations like the Kite and Fish Bases, which develop pointed structures. The progression continues to the Bird Base, which combines multiple folding techniques to create more intricate models with distinct appendages. Each base builds upon previous folding techniques, demonstrating an incremental approach to increasing geometric complexity and model potential. |
| 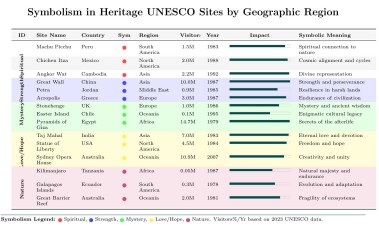 Symbolism in Heritage UNESCO Sites by Geographic Region | Which UNESCO site has the highest cultural impact indicator, and what symbolic meaning does it represent? How does this relate to its visitor numbers and year of inscription? | The Great Wall of China has the highest cultural impact indicator with a value of 120. It symbolizes 'Strength and perseverance' (as indicated by the blue color coding and symbolism type 2). Despite having the second-highest visitor numbers (10 million/year), it achieved maximum cultural impact. This suggests that while visitor numbers contribute to cultural impact, other factors like historical significance and symbolic representation play a stronger role. The site was inscribed in 1987, earlier than many others, giving it more time to establish its cultural influence. |