# OpenReview forum: "Modality-Swap Distillation: Rendering Textual Reasoning into Visual Supervision"
_ICLR.cc/2026/Conference — ICLR 2026 Conference Desk Rejected Submission_

### Official Review · Reviewer_7M3c · 2025-10-16

**Soundness:** 3
**Presentation:** 3
**Contribution:** 2
**Rating:** 4
**Confidence:** 4

**Summary:**

This paper introduces Visual-TableQA, a new large-scale, open-domain dataset for visual reasoning over complex tables. The authors identify a gap in existing benchmarks, which either lack visual complexity (text-based tables) or are limited in reasoning depth and diversity (simple image-based tables). To address this, they propose a novel, fully-autonomous data generation pipeline that leverages multiple LLMs in a collaborative framework. Key contributions include a new dataset, a novel generation pipeline, and extensive evaluation. The authors benchmark a wide range of open-source and proprietary VLMs, demonstrating that Visual-TableQA is a challenging dataset. Crucially, they show that fine-tuning models on Visual-TableQA leads to significant and generalizable performance improvements on other visual reasoning benchmarks like ReachQA and MATH-Vision, often closing the gap with much larger proprietary models.
The paper argues that by focusing on structural reasoning and leveraging a scalable, collaborative synthetic data pipeline, Visual-TableQA provides a valuable resource for training and evaluating the next generation of VLMs.

**Strengths:**

- Scalable data generation pipeline: The paper's primary strength is its highly original methodology for synthetic data generation.
- Demonstrated generalization and transferability: The most significant result is the strong generalization capability imparted by fine-tuning on Visual-TableQA. The finding that a model trained on these synthetic tables significantly improves its reasoning performance on chart-based (ReachQA) and math-based (MATH-Vision) benchmarks is powerful.
- Extensive evaluation: The paper includes a thorough evaluation of its dataset and the models trained on it.

**Weaknesses:**

- Limited human evaluation scale: The human evaluation covers 800 QA pairs, which is about 9% of the total dataset. While understandable given the scale, this is still a relatively small sample.

- Conceptual novelty: The data generation methodology, while effective, shares conceptual foundations with established paradigms like Self-Instruct [1] and other synthetic data generation pipelines [2].

- Limited applicability for RL: The dataset is excellently structured for SFT with its (image, question, answer, rationale) tuples. However, its suitability for popular optimization methods based on RL is less clear. RL requires a reliable and scalable reward signal to evaluate model-generated CoTs. The dataset struggles to verify the correctness of any arbitrary, free-form answer generated during RL training.

[1] Wang et al. Aligning Language Models with Self-Generated Instructions.
[2] Seed1.5-VL Technical Report

**Questions:**

See the weakness

---

### Official Review · Reviewer_wBKA · 2025-10-30

**Soundness:** 2
**Presentation:** 2
**Contribution:** 2
**Rating:** 2
**Confidence:** 5

**Summary:**

This paper introduces Visual-TableQA, a large-scale multimodal dataset for evaluating and enhancing visual reasoning over complex tabular data. Furthermore, they provide a modular, scalable, and autonomous pipeline for data generation. Specifically, they perform multi-model collaborative data generation via cross-model prompting and LLM-jury filtering. This "modality-swap" approach involves LLMs generating complex LaTeX tables and corresponding reasoning-intensive QA pairs. The LaTeX code is then rendered into images , effectively distilling textual reasoning ability into visual supervision. Empirical results demonstrate that open-source models fine-tuned on this synthetic dataset generalize robustly, achieving significant performance gains on external, real-world benchmarks.

**Strengths:**

- The paper introduces Visual-TableQA, a novel and low-cost dataset designed for evaluating and enhancing visual reasoning over complex tabular data. Experiments show it enables more effective knowledge distillation, emphasizing "qualitative richness over quantity".
- It presents a clear, modular, scalable, and fully autonomous data generation pipeline. This process uses multi-model collaboration, including cross-model prompting for ‘inspiration’ and an LLM-jury for filtering, to generate diverse and high-quality data.

**Weaknesses:**

- The paper's definition of "Modality-Swap" is unconvincing. Simply stating that it involves "using LaTeX code as the intermediate representation for tables" is insufficient justification for such a novel term. The explanation seems to focus more on the effects and benefits of the approach (e.g. transferring textual reasoning ability) rather than providing a clear, accurate, and well-reasoned definition of the concept itself.
- Regarding the dataset's diversity and creativity, the authors attribute this to a less-guided generation pipeline with cross-model prompting and a LLM-jury filter. This claim is not well-supported and they provide no quantitative evidence showing that this pipeline yields more diverse or creative tables/QAs than human‑annotated or template‑based baselines. Furthermore, how does the pipeline guarantee that this LLM-driven generation process is not constrained or negatively impacted by the inherent biases of LLMs?
- The designed pipeline appears to be a clever engineering approach, stitching together various VLMs, LLMs, and a series of QA tasks, rather than a fundamental methodological or conceptual innovation.
- There is a direct contradiction regarding the pipeline's automation. The abstract claims the generation pipeline is "fully autonomous" , yet the data generation pipeline section (2.2) explicitly details manual intervention steps, such as when "A human reviewer then inspects the table and makes adjustments" and when "a subset of the generated tables is manually selected".
- The paper’s central contribution is not prominent. Considering the title, "MODALITY-SWAP DISTILLATION", and the specific content, it's unclear what the authors want to emphasize. The paper feels as though it is focusing more on proving the effectiveness of the Visual-TableQA dataset. The logical connection between the concept and the dataset's utility is not sufficiently explained, and the paper lacks adequate analysis and reflection on the "Modality-Swap" concept itself.
- The experimental evidence is insufficient to support the authors' claims of generalizability. In fact, Table 5 clearly indicates that after fine-tuning Qwen2.5-VL-7B-Instruct on Visual-TableQA, the model’s performance decreased across multiple tasks in the TableVQA-Bench. This, along with other limited results, demonstrates that the authors have not provided sufficient proof for the generalizable effectiveness of Visual-TableQA.
- The claim in Section 3.1 that "The fine-tuning phase for all models was limited to one epoch to ensure consistency and reduce overfitting" seems unreasonable. The authors should provide a more detailed explanation or analysis to justify why a single epoch is a valid or optimal choice.

**Questions:**

- Could you provide a precise and formal definition of Modality-Swap Distillation？
- Could you provide more quantitative evidence and controlled ablations to emphasize your core contributions?

---

### Official Review · Reviewer_Refp · 2025-10-31

**Soundness:** 2
**Presentation:** 1
**Contribution:** 2
**Rating:** 2
**Confidence:** 3

**Summary:**

This paper presents Visual-TableQA, a new large-scale dataset aimed at improving how vision-language models reason over complex table images. The authors introduce a creative modality-swap approach that uses LLMs to generate LaTeX-rendered tables so that textual reasoning can be “translated” into visual form. Their automated pipeline combines multiple language models that generate, validate, and refine both tables and reasoning-focused question–answer pairs, all for under $100. Through extensive experiments, they show that models fine-tuned on Visual-TableQA not only perform well on table reasoning tasks but also generalize strongly to other benchmarks. Overall, the paper contributes a fresh perspective on bridging text-based reasoning and visual understanding through scalable, low-cost multimodal data generation.

**Strengths:**

1. The paper’s originality lies in its proposal of a modality-swap framework, which reimagines how textual reasoning can be converted into visual supervision through LaTeX-rendered tables. This is a fresh perspective that bridges symbolic reasoning and visual understanding, offering a creative direction beyond conventional multimodal dataset design. The automated, multi-model data generation pipeline is also innovative, demonstrating how multiple LLMs can collaborate to produce complex visual reasoning data efficiently and at very low cost.

2. In terms of quality, the study is technically solid. The generation process is well-structured, integrating data synthesis, reasoning evaluation, and model benchmarking in a consistent framework. The experimental section is broad, covering both open-source and proprietary models, and the results convincingly show that training on Visual-TableQA enhances reasoning transferability across tasks.

3. For clarity, while the writing can be dense, the paper still presents its core ideas and methodology in a generally understandable way. The figures and tables help illustrate the pipeline and results, and the organization follows a logical progression from motivation to validation.

4. Finally, the significance of the work lies in its potential to influence future multimodal reasoning research. The dataset offers a scalable and reproducible benchmark for visual reasoning over structured data, and the modality-swap idea opens possibilities for reusing textual reasoning capabilities in visual domains. Even with its limitations, the paper contributes useful conceptual and practical tools to the study of vision-language learning.

**Weaknesses:**

1. The modality-swap distillation concept is not clearly differentiated from prior work like Code-as-Intermediary Translation (CIT). It’s unclear whether the novelty lies in rendering, reasoning supervision, or the multi-model pipeline. A sharper explanation of what is genuinely new would strengthen the paper’s originality.

2. The experiments, though broad, lack depth. Results are mostly quantitative with little qualitative insight—there are no clear examples showing where models improve or fail. The reliance on an LLM “jury” for validation is also weakly justified, as the paper does not show how well these judgments align with human evaluation.

3. The writing is dense and sometimes confusing. Terms like “cross-model inspiration” and “modality-swap” are introduced without clear definitions, and the long pipeline description overshadows the main ideas. I think more intuition and motivation should be provided before the method part. More concise explanations and better figure integration would help clarity. High-quality examples will help readers to understand.

4. While the dataset is large and inexpensive to produce, its real-world usefulness is uncertain. Since all data are synthetic, improvements may reflect adaptation to artificial patterns rather than genuine progress in visual reasoning. Including human-verified or real-world experiments would make the conclusions more convincing.

**Questions:**

NA

---

### Official Review · Reviewer_TNZa · 2025-11-07

**Soundness:** 2
**Presentation:** 2
**Contribution:** 2
**Rating:** 4
**Confidence:** 3

**Summary:**

The paper addresses a gap in visual reasoning over table images and proposes an LLM-driven, LaTeX-to-image synthetic pipeline that culminates in Visual-TableQA. While the system is competently engineered, I assess that the contribution is incremental: modality-swap (textual reasoning rendered into visual supervision) closely parallels prior code-as-intermediary/synthetic VQA pipelines, with cross-model inspiration amounting to a prompt-level variation rather than a new algorithmic principle. The dataset scale is modest for current multimodal standards, and the paper does not provide rigorous evidence of statistical diversity or scalability beyond this size.
Empirically, the work reports transfer gains and an image-vs-LaTeX performance gap that supports the claim that the image modality imposes additional difficulty; however, the evaluation is confounded by (i) reliance on an LLM jury for correctness judgments, (ii) use of relaxed accuracy (explicitly acknowledged to inflate scores) without thorough sensitivity analysis, and (iii) limited ablations isolating the incremental value of the inspiration stage and jury filtering.

**Strengths:**

- A fully automated LaTeX→image pipeline, showing a practical path to bootstrapping multimodal supervision with minimal budget, though scale remains modest for current VQA norms.

- Fine-tuning on Visual-TableQA yields measurable gains and highlights a consistent image vs. LaTeX performance gap, supporting the claim that visual formatting adds difficulty; however, effect sizes rely on jury-scored accuracy and relaxed metrics, which may inflate results.

- The multi-LLM design is a competent systems contribution that likely improves diversity and quality control, even if the underlying idea parallels prior CIT-style pipelines rather than introducing a new algorithmic core.

- Public release of code/data and pipeline prompts (as claimed) enhances inspectability and reuse; this materially increases community value despite questions about dependence on closed-model components.

**Weaknesses:**

- Limited novelty. “Modality-swap distillation” and the multi-LLM pipeline largely restate known synthetic VQA/CIT patterns; “cross-model inspiration” reads as a prompt-engineering variant rather than a new algorithmic contribution.

- Small scale vs. claims of “large-scale.” The dataset is ~2.5k tables/9k QA—modest by current multimodal standards—without quantitative evidence of topic/layout diversity or demonstrated scaling beyond this size.

- Reported gains rely on an LLM jury and relaxed accuracy (acknowledged to inflate scores), with no sensitivity analysis and no ablations isolating the impact of the “inspiration” stage or jury filtering.

- Generation depends on closed models (e.g., GPT-o3/GPT-4o) raising portability concerns; comparisons omit simpler alternatives (e.g., text-only table QA baselines, single-LLM pipeline), limiting attribution of where gains come from.

**Questions:**

- what algorithmic component (beyond prompt templating) is unique, and why is “cross-model inspiration” not just prompt-level data augmentation?


- Provide an end-to-end schematic with objectives.

- Show a fully open-source reproduction (e.g., Qwen/Mistral stack) with matched hyperparams and report deltas vs. the closed-model pipeline.

- Provide: (i) text-only supervision baseline (LaTeX/CIT variant) to isolate visual gains; (ii) single-LLM vs. multi-LLM generation ablation; (iii) jury/no-jury sensitivity; (iv) comparisons to training on ReachQA/other synthetic sets under identical budgets.

---

### Note · Program_Chairs · 2026-01-17
**Submission Desk Rejected by Program Chairs**

The following references in this submission do not refer to real documents and/or have major errors in bibliographic information:

 Shaohan Chen, Yujia Zhang, Xiangpeng Cao, Shaolei He, Chen Zhao, Zhihua Liu, Chongming Li, Jing Liu, Qiang Liu, Fan Liu, et al. Qwen-vl: A versatile vision-language model with image, text, and box comprehension. arXiv preprint arXiv:2403.18751, 2024a. URL https://arxiv. org/abs/2403.18751.